# Enhanced Immunogenicity of Engineered HER2 Antigens Potentiates Antitumor Immune Responses

**DOI:** 10.3390/vaccines8030403

**Published:** 2020-07-22

**Authors:** Insu Jeon, Jeong-Mi Lee, Kwang-Soo Shin, Taeseung Kang, Myung Hwan Park, Hyungseok Seo, Boyeong Song, Choong-Hyun Koh, Jeongwon Choi, Young Kee Shin, Byung-Seok Kim, Chang-Yuil Kang

**Affiliations:** 1Department of Molecular Medicine and Biopharmaceutical Sciences, Graduate School of Convergence Science and Technology, Seoul National University, Seoul 08826, Korea; astersim@snu.ac.kr (I.J.); ksms1541@naver.com (K.-S.S.); kangts963@naver.com (T.K.); angelouss@naver.com (M.H.P.); hyungseokseo@yahoo.com (H.S.); boyeong.song@gmail.com (B.S.); garden611@naver.com (J.C.); ykeeshin@snu.ac.kr (Y.K.S.); 2Laboratory of Immunology, Research Institute of Pharmaceutical Sciences, College of Pharmacy Seoul National University, Seoul 08826, Korea; cipe@naver.com (J.-M.L.); kohch1990@snu.ac.kr (C.-H.K.); 3Division of Life Sciences, College of Life Sciences and Bioengineering, Incheon National University, Incheon 22012, Korea; byungseokkim@inu.ac.kr; 4Cellid, Inc., Seoul 08826, Korea

**Keywords:** HER2, p95HER2, cancer vaccine

## Abstract

For cancer vaccines, the selection of optimal tumor-associated antigens (TAAs) that can maximize the immunogenicity of the vaccine without causing unwanted adverse effects is challenging. In this study, we developed two engineered Human epidermal growth factor receptor 2 (HER2) antigens, K965 and K1117, and compared their immunogenicity to a previously reported truncated HER2 antigen, K684, within a B cell and monocyte-based vaccine (BVAC). We found that BVAC-K965 and BVAC-K1117 induced comparable antigen-specific antibody responses and antigen-specific T cell responses to BVAC-K684. Interestingly, BVAC-K1117 induced more potent antitumor activity than the other vaccines in murine CT26-HER2 tumor models. In addition, BVAC-K1117 showed enhanced antitumor effects against truncated p95HER2-expressing CT26 tumors compared to BVAC-K965 and BVAC-K684 based on the survival analysis by inducing T cell responses against intracellular domain (ICD) epitopes. The increased ICD epitope-specific T cell responses induced by BVAC-K1117 compared to BVAC-K965 and BVAC-K684 were recapitulated in human leukocyte antigen (HLA)-untyped human PBMCs and HLA-A*0201 PBMCs. Furthermore, we also observed synergistic antitumor effects between BVAC-K1117 and anti-PD-L1 antibody treatment against CT26-HER2 tumors. Collectively, our findings demonstrate that inclusion of a sufficient number of ICD epitopes of HER2 in cellular vaccines can improve the antitumor activity of the vaccine and provide a way to optimize the efficacy of anticancer cellular vaccines targeting HER2.

## 1. Introduction

The vaccine platform using dendritic cells (DCs) as antigen-presenting cells (APCs) has been developed and well established for the treatment of cancer in both preclinical and clinical studies [1,2,3,4,5]. However, due to the limited supply of DCs in peripheral blood [6] and the difficulty of ex vivo expansion [3], there is an urgent need to develop an alternative to DCs for APC-based cancer vaccines. In this regard, we developed a novel cancer vaccine platform, BVAC, that includes B cells and monocytes as an alternative APC source [7,8,9,10,11]. Compared to DCs, B cells and monocytes are abundant in the peripheral blood and are easy to manipulate. Although B cells and monocytes are less efficient as APCs than DCs [7,9], the addition of the natural killer T (NKT) cell ligand α-galactosylceramide (αGC) improved the function of B cells and monocytes as APCs through the bidirectional activation between NKT cells and B cells/monocytes. Moreover, introducing tumor-associated antigens through adenoviral vector systems further improved the APC function of BVAC [11].

When developing therapeutic cancer vaccines, selecting the optimal form of tumor-associated antigens (TAAs), ranging from a single precise epitope to a full-length antigen, is crucial to elicit potent antitumor activity without causing unwanted tissue damage [12,13,14]. One of the advantages of full-length antigens is epitope diversity. The wide epitope coverage provided by full-length antigen could activate not only CD8^+^ T cells but also CD4^+^ T cells. In addition, full-length antigens are relatively free from issues regarding human leukocyte antigen (HLA) haplotype matches of individual patients. Furthermore, vaccines containing multiple epitope peptides or whole antigens showed enhanced therapeutic efficacy in various animal studies compared with single epitope peptides [12,15,16]. Thus, extending the diversity of epitopes is potentially beneficial to the therapeutic efficacy of cancer vaccines.

Human epidermal growth factor receptor 2 (HER2) is one of the best-characterized TAAs in various tumors, including breast, ovarian, and gastric cancers [17,18]. HER2 has a highly oncogenic property due to its continuous signal transduction via mitogen-activated protein kinases (MAPK) and the phosphoinositide 3-kinase (PI3K)-activated Akt pathways [19]. In addition, among the naturally occurring noncanonical forms of the HER2 antigen, p95HER2, which lacks a large portion of the extracellular domain (ECD), has been associated with tumor relapse, metastasis, and the acquisition of resistance to therapeutic antibodies [20,21,22,23]. As the expression of p95HER2 is mostly restricted to tumor tissues [24,25,26], diverting immune responses toward p95HER2 might be a promising strategy to treat p95HER2-positive cancer patients. In our previous study, BVAC with a truncated form of HER2 antigen spanning a 684-amino acid length (K684) induced HER2-specific T cell responses [11]. However, since K684 only includes the extracellular domain (ECD) and transmembrane domain (TM) and the kinase domain in the intracellular domain (ICD) region has oncogenic potential [27,28], including an additional ICD region with exclusion or inactivation of the kinase domain is crucial to enable targeting p95HER2.

To introduce ICD in the BVAC without activating the kinase domain, we developed two novel BVACs, BVAC-K965 and BVAC-K1117, containing engineered forms of HER2 antigens, and compared their immunogenicity and therapeutic efficacy with the previously developed BVAC-K684. All three vaccines induced comparable humoral and T cell responses against the ECD. Interestingly, vaccination with BVAC-K1117 had increased antitumor effects against HER2- or p95HER2-expressing CT26 tumors than the other vaccines by inducing T cell responses against epitopes located in the ICD region. Furthermore, by using human PBMCs from HLA-undefined or HLA-A*0201 donors, we observed that BVAC-K1117 induced superior T cell responses compared to the other formulations. Finally, we also found that the combination of BVAC-K1117 and anti-PD-L1 antibody showed enhanced inhibition of tumor growth compared to each monotherapy. Overall, our study provides a novel strategy to optimize the efficacy of cancer vaccines targeting HER2.

## 2. Materials and Methods

### 2.1. Mice

Female 6-week-old BALB/c mice were purchased from Charles River Laboratories (Seoul, Korea). All mice were housed in specific pathogen-free conditions in the Animal Center for Pharmaceutical Research at Seoul National University (Seoul, Korea). The experiments were approved by the International Animal Care and Use Committee (IACUC) of Seoul National University.

### 2.2. Human Samples

Human peripheral blood mononuclear cells (PBMCs) were obtained from healthy donors in compliance with Institutional Review Board protocols, and informed consent was granted by all donors. The collection of human samples and all human experiments were performed in accordance with the principles of the Helsinki Declaration and approved by the ethical committee of Seoul National University and Seoul National University Bundang Hospital (IRB No. 1712/001-003).

### 2.3. Tumor Cell Lines

CT26/HER2, CT26/p95HER2, and WEHI-164 cells were cultured in DF10 medium that was DMEM media supplemented with 10% FBS and 1% penicillin-streptomycin. Human monocytic cell line THP-1 was cultured in RPMI media supplemented with 10% FBS and 1% penicillin-streptomycin. HER2-expressing CT26/HER2 cells were previously developed by transduction of CT26 cells with cDNA encoding human HER2 [29]. Briefly, CT26 wild-type cell line were transduced with MoMuLV-based retroviral vector, including the cDNA for HER2/neu and the neomycin resistance gene under the control of the SV40 promoter. The p95HER2 was amplified with polymerase chain reaction using specific primers (p95HER2for: 5′-ACTAGTAAACTACCCCAAGCTGGCCTCTGAGGCCACCATGC CCATCTGGAAGTTT-3′, and p95HER2rev: 5′-ACTAGTTTGATCCCCAAGCTTGGCCTGACA GGCCTCACACTGGCACGTCCAGACC-3′) and inserted into pSBbi-Hyg. The cloned vector pSBbi-p95HER2-Hyg were transduced into CT26 wild-type cell line with pCMV(CAT)T7-SB100 for integration of SB transposons at TA-dinucleotides of genomic DNA [30]. CT26/HER2 cells and CT26/p95HER2 cells were subcloned and maintained in DF10 supplemented with 200 μg/mL Geneticin (Gibco), and 100 μg/mL hygromycin B (InvivoGen), respectively. Tumor cell lines were validated by morphology, growth kinetics, and antigen expression.

For in vivo transplantation, 2 × 10^5^ CT26/HER2 or CT26/p95HER2 cells per mouse were subcutaneously (s.c.) injected into the left flanks of female BALB/c mice. For in vivo survival analysis, 5 × 10^5^ or 1 × 10^6^ cells of CT26/HER2 or 5 × 10^5^ cells of CT26/p95HER2 cells per mouse were injected into the tail vein. For combination therapy, 5 × 10^5^ CT26/HER2 cells were used, and monoclonal anti-PD-L1 antibody was used 5 days after BVAC administration. The rates of survival and tumor growth was measured using a metric caliper three times a week. Tumor volume was calculated as 0.5236 × length × width × height.

### 2.4. Reagent and Antibodies

The fluorochrome-conjugated antibodies to mouse B220 FITC (RA3-6B2), mouse PD-L1 FITC (10F.9G2), mouse CD3 PE (17A2), mouse CD11b PE (M1/70), mouse CD8 APC (53–6.7), human HER2 APC (24D2), mouse IgG1 PE (RMG1-1), human CD20 FITC (2H7), and human CD14 PE (M5E2) were purchased from BioLegend. FITC conjugated anti-mouse PD-1 (J43) and PE/Cy7 conjugated anti-mouse interferon-γ (XMG1.2) were purchased from Invitrogen and fixable viability dye eFluor 450 and eFluor 780 were purchased from eBioscience. The antibodies to mouse CD45.2 BV786 (104) were purchased from BD Bioscience. All microbeads for B cell and monocyte isolation (anti-mouse B220, anti-mouse CD11b, anti-human CD3, anti-human CD19, and anti-human CD14) were purchased from Miltenyi Biotec. The monoclonal anti-PD-L1 antibody (MIH5) was generated as described previously [31,32] and prepared from the ascites of nude mice by using caprylic acid purification. ICD peptide pools produced as 15-mers with 8 aa overlap and 9-mer P63 peptides (TYLPTNASL) were purchased from Anygen (Gwangju, Korea) for in vivo and in vitro cytotoxicity, and P396 (KIFGSLAFL), P435 (ILHNGAYSL), P689 (RLLQETELV), and P971 (LQRYSEDPT) for human ELISPOT assay were purchased from Cosmogenetech (Daejeon, Korea). Alpha-galactosylceramide was purchased from Enzo Life Science.

### 2.5. Generation of Modified HER2 Antigens and Adenoviral Vectors

Modified HER2 antigens were generated by PCR from HER2 wild-type (GeneID 2064) DNA. To generate modified HER2 antigens, HER2 antigens were amplified by specific primers (HF1: 5′-TTGAGTCGACATGGAGCTGGCGGCCTTGT-3′, HF2: 5′-AAGCTTCACACTGGCACGTCCAGAC-3′, HF3: 5′-CATATGAGGTGTCAGCGGCTCCACC-3, HF4: 5′-CATATGGGCC CAGCCAGTCCCT T T-3′, HF5: 5′-TTGACATATGGGAGCCCACACCAGCC-3′, HF6: 5′-CATATGGCCAAACCTTACGA TGGGATCCC-3′) and using HER2-WT plasmid as a template, which was gifted from Mien-Chie Hung (Addgene plasmid # 16257; http://n2t.net/addgene:16257; RRID: Addgene_16257) [33]. The pairs of primers were the front (HF1 to HF4) and the back (HF3 to HF2) for K965 and the front (HF1 to HF6) and the back (HF5 to HF2) for K1117. All amplified PCR products were inserted into TA cloning vectors (Biofact) and ligated after NdeI and HindIII restriction enzyme digestion. TA cloning vectors carrying HER2 antigens were digested by SalI and HindIII, and the fragmented HER2 antigens were inserted into the same restriction enzyme site of the pShuttle-CMV-tet10 (Genexine) vector. The generation of adenoviral vectors carrying modified HER2 antigens was conducted by inducing homologous recombination with pShuttle-CMV-Ag-tet10 in the *Escherichia. coli* strain BJ5183 (Agilent) carrying E1/E3-deleted and modified AdK35Easy adenoviral vector plasmid. These recombinant plasmids were transfected into human embryonic kidney 293 cells to generate virus particles. The purification of amplified adenoviruses was conducted with an AdenoX maxi purification kit (TAKARA) or FPLC with column affinity chromatography.

### 2.6. Preparation of BVAC

Splenocytes were isolated from BALB/c mice. After eliminating RBCs using ACK lysing buffer (Gibco), B220^+^ cells were isolated using anti-B220 MACS beads. After B220^+^ cells were purified, CD11b^+^ cells were isolated using anti-CD11b MACS beads. Isolated B220^+^ cells and CD11b^+^ cells were transduced with adenoviral vectors at the indicated multiplicity of infection (MOI) by centrifugation for 90 min at 2000 r.p.m at room temperature, and the cells were subsequently supplemented with 1 μg/mL αGC and incubated for an additional 18 h. After incubation, BVAC was injected into naïve or tumor-bearing mice via the tail vein.

### 2.7. Titration of HER2-Specific Antibodies

For titration of HER2-specific antibodies, CT26/HER2 cells were opsonized with serially diluted sera from immunized mice and washed with PBS. After opsonization, CT26/HER2 cells were stained with PE-conjugated anti-mouse IgG1 antibody (RMG1-1) and analyzed using flow cytometry.

### 2.8. In Vivo and In Vitro Cytotoxicity Assay

For the in vivo cytotoxicity assay, mice were vaccinated with BVAC with 2 × 10^6^ cells. Seven days after vaccination, target cells were prepared from naïve splenocytes loaded with 1 μg/mL target peptides and subsequently labeled with 5 μM CFSE (Invitrogen). Syngeneic splenocytes labeled with 0.5 μM CFSE were used as an internal control. Equal amounts of target cells and internal control cells were injected intravenously (*i.v.*) into BVAC-treated mice. The mice were sacrificed at 18 h post target cell injection, and epitope-specific target cell lysis was analyzed by FACS. The specific cell lysis was calculated as follows: r (ratio) = (% CFSE^low^/% CFSE^high^), % lysis = [1−(r^unprimed^/r^primed^)] × 100.

For the in vitro cytotoxic assay, mice were vaccinated with BVAC with 2 × 10^6^ cells. Seven days after vaccination, 2 × 10^7^ splenocytes were cocultured with mitomycin-C (Sigma-Aldrich)-treated 5 × 10^5^ CT26/HER2 or CT26/p95HER2 cells. After 5 days, CD8^+^ T cells were isolated from cocultured splenocytes with anti-CD8 MACS beads. The target cells were WEHI-164 cells that were loaded with target epitopes (2 μg/mL) for 4 h and subsequently labeled with ^51^Cr. Target epitope-loaded ^51^Cr-labeled WEHI-164 tumor cells were cocultured with CD8^+^ T cells for 4 h. The specific cell lysis was analyzed by Wallac 1470 Wizard automatic γ-counter (Perkin Elmer) measuring ^51^Cr in the supernatant. Specific lysis (%) was calculated as follows:% lysis = [(experimental release-spontaneous release)/(maximum release–spontaneous release)] × 100(1)

### 2.9. Immunospot Analysis

For immunospot analysis, 1 × 10^6^ αGC-loaded adenovirus-transduced human CD3^-^ PBMCs were cocultured with 1 × 10^8^ autologous PBMCs in chemically defined X-vivo 15 (Lonza) media. The media was substituted with media supplemented with IL-2 (5 ng/mL) and IL-15 (5 ng/mL) on days 3 and 7. At the end of coculture, CD8^+^ and CD4^+^ T cells were isolated from cocultured PBMCs using magnetic beads and seeded on human interferon-gamma (IFN-γ) ELISPOT plates (Cellular Technology Ltd., Cleveland, OH, USA) for the indicated peptide restimulation.

### 2.10. Statistics

The data are shown as the means ± s.e.m., and statistical significance was analyzed with GraphPad Prism software. A two-tailed Student’s *t*-test was used to compare differences between two groups. Both one-way and two-way analysis of variance were used to compare differences among multiple groups. *p*-values < 0.05 were considered statistically significant.

## 3. Results

### 3.1. Development of Adenoviral Vectors Coding for Engineered HER2 Antigens

To address whether insertion of the ICD region can improve the immunogenicity of the HER2 antigen, we designed two novel HER2 antigens that contain the ICD region. Since the tyrosine kinase domain in the ICD region (a.a. 720–976) has been associated with oncogenic potential, it was disrupted by removing a.a. 703–994 or a.a. 780–919 [34], resulting in K965 or K1117, which were named based on the total antigen length (Figure 1A,B). The adenoviral vectors (Ad) carrying the engineered HER2 antigens were constructed by restriction enzyme cloning and homologous recombination using the BJ5183 *E. coli* strain [35]. Then, the virus particles produced and amplified in the HEK293R cell line were purified by affinity chromatography. To evaluate the vector-induced expression of HER2, we transduced AdK684, AdK965, or AdK1117 an MOI of 100 into THP-1 human monocytic cells. After 24 h, the expression of HER2 was analyzed by flow cytometry (Figure 1C). All three vectors efficiently induced HER2 expression in THP-1 cells, while the mean fluorescence intensity (MFI) of HER2 expression in AdK684 cells was higher than that in the other two cells within the live population (Appendix A). Consistent with the results observed in THP-1 cells, adenoviral vectors containing the engineered HER2 antigens efficiently induced HER2 expression in primary human CD20^+^ B cells and CD14+ monocytes and murine B cells (Figure 1D,E and Appendix A). In addition, we tested the attenuation of HER2 signaling by analysis of phosphorylated Erk. We transduced THP-1 cells with indicated adenovirus at an MOI of 20, and the cytoplasmic proteins were analyzed by immunoblot 24 h after virus transduction. As a result, each adenovirus transduced THP-1 cell line showed attenuation of the phosphorylation of Erk compared to the AdHER2WT cell line (Appendix A).

### 3.2. Immunization with the Engineered Antigens Elicits HER2-Specific Humoral and Cellular Immune Responses

Next, we tested whether the engineered antigens induce HER2-specific immune responses. To this end, we transduced each engineered HER2 antigen-expressing vector into NKT cell ligand-loaded B cell and monocyte vaccine (BVAC) as previously demonstrated. After immunizing BALB/c mice with each BVAC expressing a different form of the HER2 antigen, HER2-specific antibody titers in the sera at different time points after immunization were determined by a binding assay using HER2-expressing CT26 (CT26/HER2) tumor cells (Figure 2A,B). We observed a gradual increase in the titers of HER2-specific antibodies in all three tested HER2-expressing BVAC immunization groups. By contrast, vector transduction of BVAC without HER2 (Bmo/αGC) did not induce HER2-specific antibody responses, suggesting that the antibody responses were induced by the HER2 antigens expressed on BVAC. The antibody titers peaked approximately 9–11 weeks after immunization and then decreased thereafter. Of note, BVAC-K1117 was less efficient in inducing antibody responses than the other two vaccines.

Using the H2-Kd binding P63 peptide, which is located in the ECD region [36], we examined the cellular immune responses of each engineered antigen against HER2. To determine P63-specific cytotoxicity, splenocytes isolated from immunized mice were stimulated with the mitomycin-C-treated CT26/HER2 tumor cell line for 5 days and then subjected to a ^51^Cr release assay using P63-specific peptides (Figure 2C). All three BVACs expressing HER2 antigen induced potent target cell killing activity in comparison to BVAC without HER2 expression and BVAC-K965 was the most potent. In addition, to assess the cytotoxic effect in vivo, we immunized mice with each BVAC and then injected CFSE^hi^-labeled P63 peptide-loaded splenocytes together with CFSE^lo^-labeled peptide-unloaded control splenocytes 1 week after immunization. Similar to in vitro analysis, all three BVACs expressing HER2 antigen induced potent target cell killing activity in vivo compared to BVAC without HER2 expression but BVAC-K965 was the most potent (Figure 2D). Collectively, these results suggest that HER2 antigens containing the ICD region without the tyrosine kinase domain can induce potent humoral and cytotoxic immune responses.

### 3.3. BVAC with the K1117 Antigen Efficiently Inhibits the Growth of HER2-Expressing Tumors

We next tested the antitumor effect of BVACs expressing engineered HER2 antigens in mouse tumor models. Naïve BALB/c mice were inoculated with CT26/HER2 tumor cells and then treated with each BVAC at day 6 (Figure 3A). Consistent with the comparable antibody titers and CTL responses, BVAC-K965 inhibited tumor growth as efficiently as BVAC-K684. Unexpectedly, we observed that the antitumor effect of BVAC-K1117 was indistinguishable from that of BVAC-K684 and BVAC-K965 despite the relatively poor capacity to induce antibody responses or cytotoxic responses (Figure 2). The tendency of the antitumor effect of engineered antigens was found in the survival analysis (Figure 3B). To further evaluate the therapeutic effect of BVACs expressing engineered HER2 antigens, we tested a multiple injection regimen in the same CT26/HER2 tumor model (Figure 3C). Intriguingly, we observed a significant delay in tumor growth with BVAC-K1117 vaccination compared to BVAC-K684 or BVAC-K965 vaccination. Notably, as dead cells could be an immune stimulator causing incorrect interpretation of the observed results, we analyzed the live and dead cell populations of BVAC. The percentage of dead BVAC-K1117 cells was significantly lower than that of the K684 and Bmo/αGC (Appendix A). These results indicated that dead BVAC cells did not affect antitumor responses. Taken together, these results suggest that the engineered antigen K1117 would be better than K684 and K965 at inducing antitumor immune responses.

### 3.4. BVAC-K1117 Induces Antigen-Specific Cytotoxic Responses Against ICD of HER2

As BVAC-K1117 showed the most potent antitumor effect among the three BVACs, we hypothesized that the epitopes located in the ICD region of K1117 played a crucial role in eliciting superior antitumor responses. When the 15-mer ICD epitope peptide pool-loaded target cells were adoptively transferred into vaccinated mice, BVAC-K1117 showed a higher target killing effect than BVAC-K684 and BVAC-K965 (Figure 4A). To compare the antitumor response against the ICD region of HER2, we established a p95HER2-expressing CT26 cell line (CT26/p95HER2) that lacks the majority of the ECD region of HER2, including the P63 epitope. We inoculated CT26/p95HER2 cells into naïve BALB/c mice and then vaccinated the mice with each BVAC on day 6. Compared to control BVAC (Bmo/αGC), BVAC-K965 and BVAC-K1117 significantly delayed tumor growth, whereas tumor regression was minimal in the BVAC-K684 group (Figure 4B). However, in contrast to the CT26/p95HER2 tumor growth of subcutaneous models, the survival of intravenously inoculated CT26/p95HER2 bearing mice indicated that BVAC-K1117 is superior to BVAC with other antigens (Figure 4C). These data were consistent with the target killing effect against the ICD region of HER2 (Figure 4A). In addition, we sought to investigate whether supplementation with ICD epitope peptides can improve the antitumor effect of BVAC-K684 against CT26/p95HER2 tumors. As we expected, BVAC-K684 pulsed with ICD peptide pools (BVAC-K684^ICD^) elicited a significantly enhanced therapeutic effect against CT26/p95HER2 tumors compared to BVAC-K684 alone (Figure 4D). Collectively, these results suggest that inclusion of the ICD region of HER2 improves the antitumor effect of BVAC by promoting ICD-specific immune responses.

### 3.5. Engineered HER2 Antigens Enhance HER2-Specific T Cell Responses in Human PBMC Models

To extend our findings to the clinical application of HER2 targeting BVAC, we investigated the immunogenicity of engineered HER2 antigens in HLA un-typed human PBMCs. We cocultured PBMCs with autologous BVACs expressing each engineered HER2 antigen under IL-2 plus IL-15 conditions (Figure 5A). After 7 days of culture, we sorted CD8^+^ and CD4^+^ cells from cocultured PBMCs and restimulated them with CD3-depleted PBMCs loaded with pooled HER2 epitope peptides. Interestingly, we found that human PBMCs primed with BVAC-K1117 induced the highest immune responses, as estimated by the increased number of IFN-γ^+^ spots from cocultured CD8^+^ as well as CD4^+^ T cells (Figure 5B–D).

We next tested whether BVAC with engineered HER2 antigens could elicit immune responses specific to MHC class I-restricted epitopes located either in the ECD or ICD. We obtained peripheral blood from healthy donor carrying HLA-A*0201, and the PBMCs were cocultured with autologous BVAC. As depicted in Figure 5E, CD8^+^ T cells and CD4^+^ T cells sorted from cocultured PBMCs were restimulated with peptide-loaded autologous CD3^-^ PBMCs 11 days after coculture. We used HLA-A2-restricted ECD 9-mer peptides (P369 and P435) and ICD 9-mer peptides (P689 and P971) for CD8^+^ T cell restimulation [37,38] and pooled 15-mer ECD peptides or pooled 15-mer ICD peptides for CD4^+^ T cell restimulation. Notably, BVAC-K1117 induced more HER2-specific IFN-γ^+^ CD8^+^ T cells against ICD peptides and ECD peptides than BVAC-K684 and BVAC-K965 (Figure 5F,G). In addition, BVAC-K1117 was superior to BVAC-K684 and BVAC-K965 at inducing ICD-specific IFN-γ production by CD4^+^ T cells, although it could not increase the number of ECD-specific IFN-γ^+^ CD4^+^ T cells compared to the control (Figure 5H,I). Collectively, these results suggest that BVAC-K1117 can efficiently induce ICD-specific CD4^+^ and CD8^+^ T cell responses in HLA-defined and HLA-undefined PBMCs.

### 3.6. The Combination of Anti-PD-L1 Therapy with BVAC-K1117 Augments Antitumor Immunity in Murine Tumor Models

Therapies targeting the PD-1/PD-L1 pathway, such as anti-PD-1 and anti-PD-L1 antibodies, have proven remarkable therapeutic efficacy against cancer in the clinic. Thus, we tested whether BVAC-K1117 can synergistically enhance the therapeutic efficacy of anti-PD-L1 antibody in a CT26/HER2 tumor model. As shown in Figure 6A, BALB/c mice inoculated with CT26/HER2 cells were treated starting from day 5 after tumor injection. After multiple treatments with either BVAC or anti-PD-L1 antibody, tumor growth was mitigated compared to that in control IgG-treated mice (Figure 6B). Since the vaccination with BVAC-K1117 dramatically induced IFN-γ expression on tumor-infiltrating CD8 T cells (Appendix A) and the IFN-γ produced by tumor-infiltrating CD8 T cells is a potential inducer of PD-L1 expression on tumor cells [39], we hypothesized that IFN-γ-dependent induction of PD-L1 expression on tumor cells contributed to the additive anti-tumor effect of anti-PD-L1 in BVAC-K1117 vaccinated mice. To test this hypothesis, we examined PD-L1 expression on tumor after BVAC-K1117 vaccination. Although the IFN-γ production in tumor-infiltrating CD8 T cells was considerably increased by BVAC-K1117 vaccination compared to without vaccination or Bmo/aGC vaccination, PD-L1 expression on tumor cells was not significantly changed by the BVAC-K1117 vaccination (Appendix A), suggesting that enhanced anti-tumor effect by the addition of anti-PD-L1 was not due to IFN-γ-dependent induction of PD-L1 expression on tumor cells by BVAC-K1117 vaccination. Taken together, these results suggest that anti-PD-L1 treatment can potentiate the antitumor effect of BVAC-K1117.

## 4. Discussion

The advantages of using the whole antigen covering total antigenic epitopes rather than using the truncated antigen encompassing specific immunogenic epitopes in the cancer vaccine platform has been documented in preclinical and clinical studies [1,2,3,4,5]. However, in the case of HER2-targeting cancer vaccines, the oncogenic potential of HER2 has been one of the major impediments to whole antigen utilization. In this report, we designed two novel truncated HER2 antigens containing multiple immunogenic epitopes in the ICD region as well as in the ECD region to potentiate its immunogenicity while excluding kinase domain in the ICD region to attenuate its oncogenic potential.

Highly oncogenic signal transduction of HER2 is initiated by autophosphorylation of the cytoplasmic domain after homo or heterodimerization with other human epithelial receptor family members or estrogen receptors [28,40]. As autophosphorylation has a crucial role in oncogenic signaling, a mutated HER2 whose ATP-binding lysine residue was substituted with alanine (K753A), was developed and tested for safety and therapeutic relevance [41,42]. Although the single amino acid substitution significantly improved the therapeutic potential of the HER2 antigen without inducing oncogenic signaling, safety concerns remain presumably due to the random reactivation of kinase function [42]. Thus, it is important to ensure the irreversible inactivation of the kinase domain in the HER2 antigen to avoid potential safety concerns. In this study, we observed that inclusion of a large portion of ICD region of HER2 antigen that contains several immunogenic epitopes but not the kinase domain in the cellular cancer vaccine improved the antitumor effect of the vaccine without a significant induction of Erk phosphorylation, one of the indicators of oncogenic HER2 signaling.

Earlier studies suggested that cellular vaccination with whole antigen transduced by viral vectors results in an increased immune response compared to that with a single peptide [12,43]. As the whole protein antigen might include multiple CD4+ T cell epitopes as well as CD8^+^ T cell epitopes, overall immune responses would be enhanced by using whole antigen. In the present study, we compared the therapeutic effects of engineered antigens to demonstrate the importance of the presence of multiple epitopes. Comparison of tumor antigen-specific immune responses revealed that K1117, which contained more ICD regions than K965 and K684, elicited the superior antitumor effect and immunogenicity in mice and human PBMCs, respectively, to the other two antigens. Furthermore, supplementation with ICD epitopes dramatically improved the antitumor activity of BVAC-K684. Therefore, our data indicate that inclusion of multiple antigenic epitopes in the vaccine platform is more required to attain optimal antitumor immune responses.

Given the importance of tumor-specific antibodies in antitumor therapies [44], we hypothesized that induction of humoral responses is essential to effectively control tumor progression. Unexpectedly, however, despite the induction of potent antitumor activity, BVAC-K1117 vaccination induced less antitumor antibody responses than BVAC-K684 vaccination. Considering that ICD-specific CD8^+^ T cell responses as well as ICD-specific CD4^+^ T cell responses were higher with BVAC-K1117 vaccination than with BVAC-K684 vaccination, inducing ICD-specific T cell responses would be more important to promote antitumor responses than eliciting tumor antigen-specific antibody secretion at least in our HER2-specific tumor models. However, since tumor antigen-specific antibodies can contribute to the antitumor effect in several different ways, such as antibody-dependent cytotoxicity and phagocytosis of opsonized tumor cells [45], we do not rule out the potential involvement of HER2-specific antibodies in the antitumor effect of BVAC-K1117.

Immune checkpoint blockade targeting PD-1 or PD-L1 has been developed to reinvigorate the function of exhausted T cells and has shown successful clinical outcomes [46,47]. However, it has been reported that only a subset of patients benefited from checkpoint blockade therapy in the clinic. As the pre-existing tumor specific T cells in the tumor site might be required for the patients’ responsiveness to checkpoint blockade [48,49,50], boosting tumor antigen-specific immune responses prior to checkpoint blockade has shown clinical benefits [51]. In addition, it has been reported that the increased T cell receptor (TCR) diversity of peripheral PD-1^+^CD8^+^ T cells before treatment in patients with non-small cell lung cancer was associated with better clinical outcome in checkpoint blockade therapy [52]. Thus, induction of diverse tumor-specific CD8^+^ T cells is necessary to overcome refractoriness to checkpoint inhibitors. Here, we observed that BVAC-K1117 vaccination that can induce diverse tumor-specific CD8^+^ T cells against ICD epitopes as well as ECD epitopes of the HER2 antigen showed enhanced antitumor effects when combined with anti-PD-L1 checkpoint blockade therapy. Therefore, inclusion of multiple epitopes, including ICD-specific epitopes, in HER2 antigen might improve the efficacy of anti-HER2 cancer vaccines as a combination therapy for checkpoint blockade.

Although it has been reported that expression of HER2 in normal tissues is less significant than that in cancers [53], potential of normal tissue damage induced by anti-HER2 immunotherapy has been documented. For example, HER2 blocking antibodies combined with chemotherapy induced cardiotoxicity in metastatic breast cancer [54,55]. In addition, chimeric antigen receptor T cell (CART) therapies targeting HER2 elicited serious on-target, off-tumor adverse events [56]. In contrast, clinical trials of DC-based vaccines suggested that cellular vaccines expressing HER2 antigen could be well-tolerated without causing severe adverse effects [57]. Although we tested the antitumor effect of BVAC expressing human HER2 in human HER2-expressing mouse colon cancer tumor models, we did not observe any behavioral changes in mice after BVAC-HER2 vaccinations. Given that the degree of homology between human and mouse HER2 is extremely high (88% identical to each other) [58] and that vaccination with human HER2 antigen, K684, expressing DC vaccine induced mouse tumor models in our previous study [59], we speculate that the normal tissue damage induced by BVAC-HER2 vaccination might be minimal even in HER2^+^ cancer patients.

In summary, our study demonstrated that engineering HER2 antigens in a cellular vaccine platform to include a majority of the ICD region except the tyrosine kinase domain improved the antitumor effect of the vaccine while relieving the safety concerns regarding the oncogenic potential of this HER2 in humans. Our findings provide insight into the development of optimal tumor antigens for cellular cancer vaccines.

## 5. Conclusions

In this study, we developed two novel HER2 antigens, K965 and K1117, that are engineered to include multiple immunogenic epitopes but not oncogenic kinase domain in the ICD region. Within a previously described B cell based cellular cancer vaccine platform, the immunogenicity and anti-tumor effect of these engineered antigens were evaluated by comparing previously described truncated HER2 antigen, K684, that contains limited immunogenic epitopes. Although BVAC-K1117 that encompasses more immunogenic epitopes located in the ICD regions than BVAC-K684 and BVAC-K965 induced relatively weak antigen-specific antibody responses, it induced more potent antitumor activity than the other two vaccines in murine CT26/HER2 tumor models. In addition, due to the T cell responses specific to ICD epitopes, BVAC-K1117 showed enhanced antitumor effects against non-canonical HER2 (p95HER2) expressing CT26 tumors compared to BVAC-K684. BVAC-K1117 also showed superior tumor antigen-specific T cell responses compared to BVAC-K684 and BVAC-K965 in human PBMCs from HLA undefined and HLA-A*0201 donors. Furthermore, enhanced antitumor effect was observed by combination of BVAC-K1117 with anti-PD-L1 antibody treatment against CT26-HER2 tumors. Collectively, the overall evaluation indicates that expression of a large portion of ICD domain of HER2 except the kinase domain in the anti-HER2 cellular cancer vaccines guarantees antitumor potency but minimizes the safety concerns of the vaccines.

## Figures and Tables

**Figure 1 vaccines-08-00403-f001:**
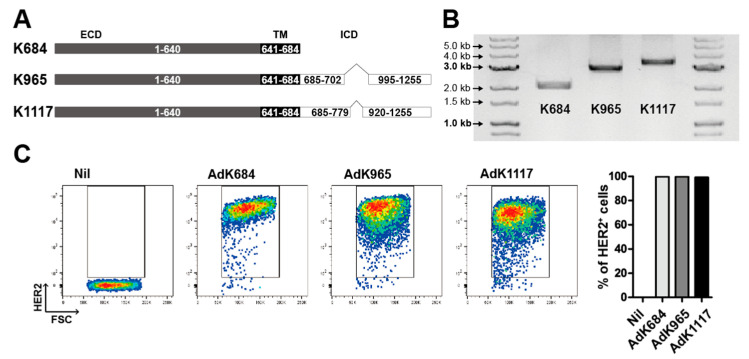
The development of three different engineered human epidermal growth factor receptor 2 (HER2) antigens. (**A**) Diagram of the concept of engineered HER antigens based on the extracellular domain, transmembrane domain, and intracellular domain. (**B**) The gel image of each antigen. The K684, K965, or K1117 (from left to right, respectively) construct was inserted into an adenovirus shuttle vector and digested with restriction enzymes SalI and HindIII. The expression of engineered HER2 antigens on the THP-1 (**C**) and CD14+ monocytes (**D**) and CD20+ B cells (**E**) isolated from human peripheral blood (**D**) was analyzed by flow cytometry. All data are representative of two independent experiments.

**Figure 2 vaccines-08-00403-f002:**
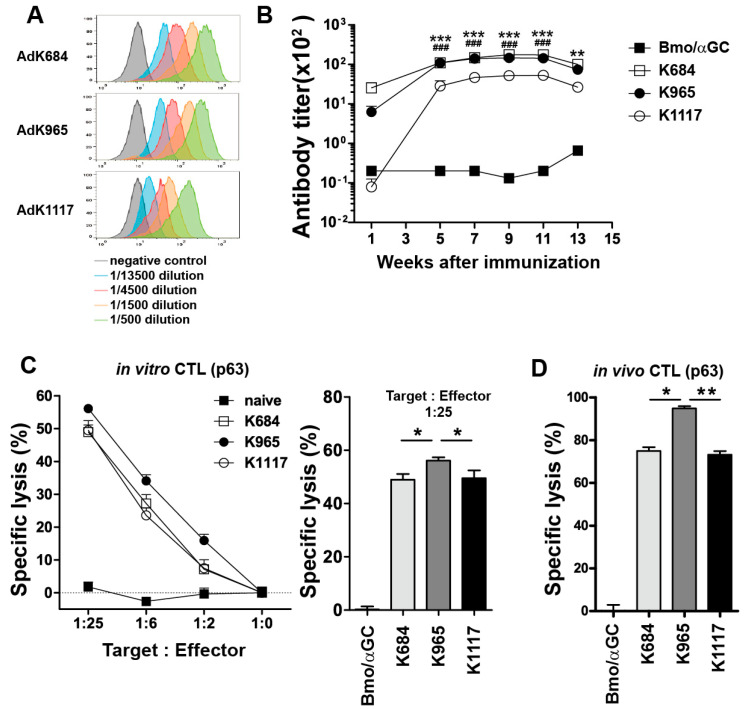
The analysis of the antibody response in the sera of vaccinated mice. (**A**,**B**) Naïve BALB/c mice were immunized with B cell and monocyte-based vaccine transduced by modified adenovirus carrying each antigen. One, 5, 7, 9, 11, and 13 weeks after immunization, antibodies in the mouse sera were titrated by flow cytometry (* *p* < 0.05, ** *p* < 0.01, *** *p* < 0.001 for K684 versus K1117, ^###^
*p* < 0.001, for K965 versus K1117). In vitro and in vivo cytotoxic T lymphocyte (CTL) analysis. Naïve BALB/c mice were immunized with B cell and monocyte-based vaccine transduced by modified adenovirus carrying each antigen. The splenocytes from immunized mice were co-cultured with attenuated CT26/HER2 cell lines on day 7. After 5 days CD8+ T cells were sorted by microbeads and cocultured with ^51^Cr-labeled P63 target peptide-loaded WEHI-164 cells (**C**). For in vivo CTL analysis, naïve BALB/c mice were immunized with B cell and monocyte-based vaccine transduced by modified adenovirus carrying each antigen. HER2 P63 peptide-loaded target cells and unloaded internal control cells labeled with CFSE high and low, respectively, were injected into immunized mice. One day after target cell injection, all mice were sacrificed, and specific cell lysis was analyzed with flow cytometry (**D**). (* *p* < 0.05, ** *p* < 0.01, *** *p* < 0.001).

**Figure 3 vaccines-08-00403-f003:**
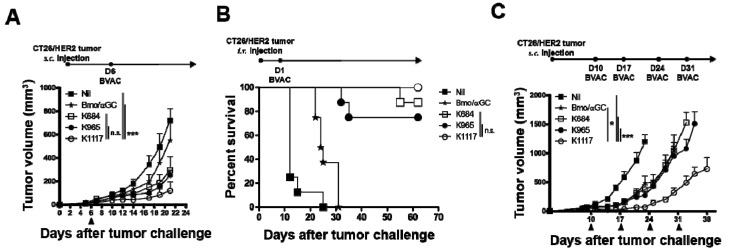
Therapeutic effects in mouse tumor models. A total of 2 × 10⁵ HER2-expressing CT26 tumor cells were subcutaneously injected into naïve BALB/c mice and immunized with B cell and monocyte-based vaccines (BVACs) carrying the indicating engineered antigens (**A**). (**B**) A total of 1 × 10^6^ CT26/HER2 tumor cells were injected into tail vein of naïve BALB/c mice. On day 3, tumor bearing mice were immunized with BVACs carrying the indicating engineered antigens and survival time was monitored. (**C**) CT26/HER2 tumor-bearing mice were administered BVACs with engineered antigens four times when the tumor size reached 80 mm^3^. All statistical analyses were compared to K1117. Subcutaneous tumor models are representative of two independent experiments (* *p* < 0.05, *** *p* < 0.001, n.s. *p* > 0.05).

**Figure 4 vaccines-08-00403-f004:**
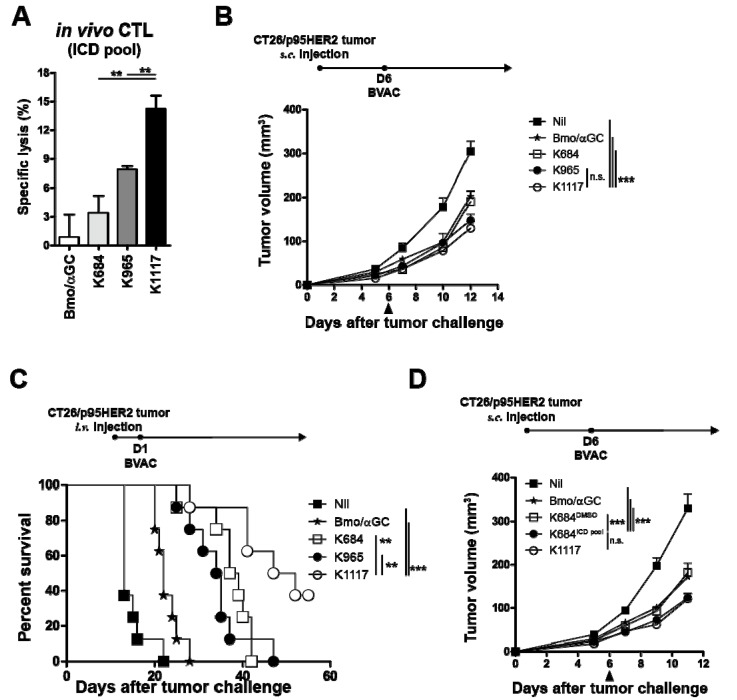
The function of epitopes located in the intracellular domain (ICD) of HER2. Naïve BALB/c mice were immunized with BVACs coding for each engineered antigen. On day 8, splenocytes from naïve BALB/c mice were pulsed with the HER2 ICD 15 mer peptide pool and labeled with CFSE^high^. Peptide loaded CFSE^high^ labeled splenocytes were injected into immunized mice with the same number of peptide unloaded CFSE^low^ labeled splenocytes as an internal control. One day after target cell injection, all mice were sacrificed, and specific cell lysis was analyzed with flow cytometry (**A**). A total of 2 × 10^5^ CT26/p95HER2 tumor cells were injected into the left flanks of naïve mice and tumor-bearing mice were immunized with BVACs as indicated (**B**). A total of 5 × 10^5^ CT26/p95HER2 tumor cells were injected into tail vein of naïve BALB/c mice. One day after tumor inoculation, tumor bearing mice were immunized with BVACs carrying the indicating engineered antigens and survival time was monitored (**C**). (**D**) The same as (**B**), 2 × 10^5^ CT26/p95HER2 were inoculated into the left flanks of naïve mice and administrated BVAC with indicated conditions. All statistical analyses were compared to K1117. All data are representative of at least three independent experiments except survival analysis (** *p* < 0.01, *** *p* < 0.001, n.s. *p* > 0.05).

**Figure 5 vaccines-08-00403-f005:**
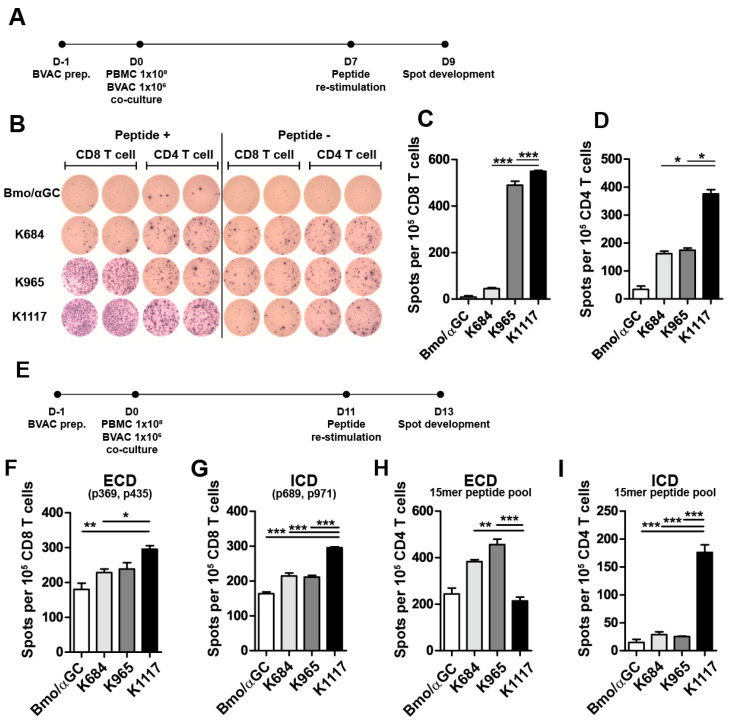
Human T cell responses against HER2. Human PBMCs from human leukocyte antigen (HLA) undefined donors were cocultured with autologous BVACs with engineered antigens (**A**). After 7 days, the CD8 and CD4 T cells were sorted from cocultured PBMCs and restimulated with a total HER2 peptide pool, and IFN-γ production was measured with an ELISPOT assay (**B**–**D**). (**E**) Human PBMCs carrying HLA-A*0201 were obtained by leukapheresis from a healthy donor and cocultured with autologous BVACs with engineered antigens. After 11 days, the CD8 T cells isolated from cocultured PBMCs were restimulated with HER2 HLA-A*0201-restricted extracellular domain (ECD) or ICD 9-mer peptides (**F**,**G**), respectively, CD4 T cells were restimulated with the ECD or ICD 15-mer peptide pool, and then IFN-γ secretion was measured with ELISPOT assay (**H**,**I**), respectively. All data are representative of three independent experiments (* *p* < 0.05, ** *p* < 0.01, *** *p* < 0.001).

**Figure 6 vaccines-08-00403-f006:**
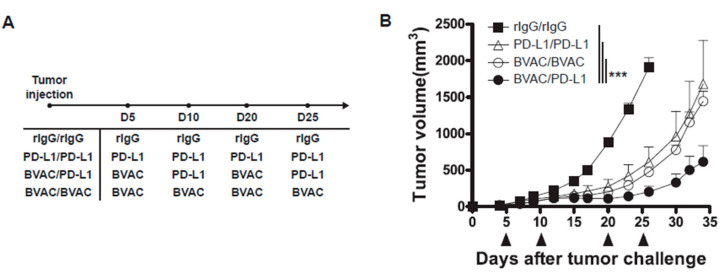
The effect of the combination of BVAC-K1117 with anti-PD-L1 antibody therapy on CT26/HER2 tumor-bearing mice. We inoculated 5 × 10^5^ CT26/HER2 cells into naïve BALB/c mice, and vaccinated the mice with the indicated therapy on days 5 and 10 for the first cycle and on days 20 and 25 for the second cycle (**A**,**B**). The dose per treatment of BVAC was 2 × 10^6^ cells per mouse and anti-PD-L1 antibody (MIH5) was 300 μg per mouse. All data are representative of two independent experiments (*** *p* < 0.001).

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
