# Peer review of "Enhanced Immunogenicity of Engineered HER2 Antigens Potentiates Antitumor Immune Responses"

_vaccines, 2020, doi:10.3390/vaccines8030403_

Round 1

Reviewer 1 Report

The authors have responded to all of the criticisms raised except one: they do not provide the ATCC certificate for the used cell lines. They have to perform the STRs CELL ID to confirm the data.

Author Response

We agree with the reviewer that it is normally required to provide the ATCC certificate for the cell lines used in the study. Unfortunately, however, the certificates for both CT26 and WEHI-164 are not currently available in our hand. Because it has been a long while since we purchased the original cell lines from the ATCC, it might have been accidentally lost during a laboratory relocation. We appreciate the reviewer for suggesting STRs CELL ID method as an alternative way to confirm the identity of the cell lines. But we regret that we will not be able to meet the submission deadline (Jul 14) before completing the test since it will take more than a month. Since we used an early passage stock of the original CT26 cell line to generate CT26-HER2 or CT26-p95HER2 cell lines, we believe that the transformed CT26 lines maintain the characteristics of the original CT26 cell line. Considering that the purpose of the current study is the optimization of HER2 antigens in the vaccine platform to treat HER2-expressing cancers in general but not the specific HER2-expressing colon cancers, we hope that the lack of the ATCC certificate does not seriously harm the conclusion of the study.

Reviewer 2 Report

All my previous concerns in this manuscript have been sufficiently addressed.

However, I did not see any figure in the associated supplementary file, hence I could not evaluate some of the statements made in the text that refers to supplementary figures. Assuming the findings in the supplementary figures tally with the text, this manuscript may be accepted for publication. 

Author Response

We appreciate the reviewer’s comments. We had initially uploaded the supplementary data as a compressed file containing supplementary figures as a PowerPoint file and figure legends/methods as Word files. As per reviewer #2’s request, we generated a single merged pdf file containing all the supplementary information.

Reviewer 3 Report

The revised version of the manuscript has been substantially improved. Please revise some misspellings in the text.

A comment on two minor errors:

The legend of Figure 6 describes putative Figure 6C, 6D, 6E. But the mentioned Figures (6C-E) are not visible in Figure 6. Of note, in the main text one finds allusions to supplementary Figure 5A-C, but not to Figure 6C-E. Could it be that the authors decided to move the figures that initially were in Fig 6C-E to Figure S5A-C?

In the legend corresponding to Supplementary Figure 5, the B and C captions seem to have been changed by D and E. Please revise.

Author Response

Thanks for bringing these errors to our attention. As the reviewer mentioned, we relocated Figure 6C-E to supplementary Figure 5, but we did not revise the figure legends accordingly. We have corrected the errors in the legend of Figure 6 and supplementary Figure 5A-C as follows.

Original.

Figure 6. The effect of the combination of BVAC-K1117 with anti-PD-L1 antibody therapy on CT26/HER2 tumor-bearing mice. We inoculated 5 х 105 CT26/HER2 cells into naïve BALB/c mice, and vaccinated the mice with the indicated therapy on days 5 and 10 for the first cycle and on days 20 and 25 for the second cycle (A and B). The dose per treatment of BVAC was 2 x 106 cells per mouse and anti-PD-L1 antibody (MIH5) was 300 μg per mouse. The expression of PD-L1 molecules on tumor and the recruitment of tumor specific CD8 T cells were analyzed. 5 x 105 cells of CT26/HER2 was inoculated into naïve BALB/c mice and BVAC was administered on day 5. Seven days after vaccination, the tumor tissues were dissociated and analyzed the expression of PD-L1 and tumor infiltrating CD8 T cells (C). The MFI of PD-L1 on tumor cells (D) and the number of tumor specific CD8 T cells per gram tumor was analyzed (E). All data are representative of two independent experiments (*P<0.05, **P<0.01, ***P<0.001).

Revised.

Figure 6. The effect of the combination of BVAC-K1117 with anti-PD-L1 antibody therapy on CT26/HER2 tumor-bearing mice. We inoculated 5 х 105 CT26/HER2 cells into naïve BALB/c mice, and vaccinated the mice with the indicated therapy on days 5 and 10 for the first cycle and on days 20 and 25 for the second cycle (A and B). The dose per treatment of BVAC was 2 x 106 cells per mouse and anti-PD-L1 antibody (MIH5) was 300 μg per mouse. All data are representative of two independent experiments (*P<0.05, **P<0.01, ***P<0.001).

Original.

The expression of PD-L1 on HER2+ CD45.2- CT26/HER2 tumor cells was analyzed (B and C). The tumor infiltrating lymphocytes were stimulated with HER2 epitope (hP63) and analyzed the secretion of IFN-g by CD8+ T cells were analyzed by flow cytometry (D and E). (*P<0.05, **P<0.01, ***P<0.001).

Revised.

The tumor infiltrating lymphocytes were stimulated with HER2 epitope (hP63) and analyzed the secretion of IFN-g by CD8+ T cells were analyzed by flow cytometry (B and C). The expression of PD-L1 on HER2+ CD45.2- CT26/HER2 tumor cells was analyzed (D and E). (*P<0.05, **P<0.01, ***P<0.001).

This manuscript is a resubmission of an earlier submission. The following is a list of the peer review reports and author responses from that submission.

Round 1

Reviewer 1 Report

The study by Jeon and colleagues is focused on the development of engineered HER2 antigens used as epitopes presented by a new platform of tumor vaccines, the BVAC, that uses B cells and monocytes as APC cells, rather than dendritic cells. The same group has already published a work on the same topic, but there in an interesting point to be discussed in the current research: the previous BVAC was loaded by a partial HER2 (called K684) essentially composed by the ECD and the TM of HER2. In the present paper, they wonder if the addiction of partial ICD can lead to an increase in  vaccine efficiency. To do this, they developed thought PCR generation and adenoviral cloning other two BVACs with HER2 engineered HER2 antigens with partial ICD, lacking the Tyrosine Kinase oncogenic domain, called K965 and K1117, respectively. Moreover, the rationale behind the research is also based on the concept that increasing the epitopes diversity can enhance the benefit derived from a tumoral vaccine therapy. To describe the features of the two new HER2 truncated antigens they:

  • Evaluate the ability of the cloned product to generate HER2 antigens
  • Evaluate the ability of the produced BVACs-HER2 to induce both humoral and cellular specific response in immunized BALB/c mice
  • Evaluate the ability of the produced BVACs-HER2 to reduce the tumor growth in BALB/c mice inoculated with CT26/HER2 tumor cells
  • Evaluate the increase in efficiency of the 2 new BVACs-HER2 with partial ICD compared with the previous, lacking this protein domain by evaluating the tumor growth in BALB/c mice inoculated with CT26/p95HER2 tumor cells, lacking the ECD domain
  • Evaluate the immunogenicity of the new BVACs-HER2 by co-culture BVACs with both un-typed and HLA-A*0201 human PBMCs
  • Evaluate the cumulative effects of a combined BVACs-HER2 and anti-PD1 therapy.

The authors, after demonstrating the ability to generate proteins by new constructs, showed both humoral and cellular specific responses, enhanced with K684 and K965 compared with K1117. All of the BVACs show a good antitumor activity, however, the K1117 seems to be more efficient in reducing tumor growth in the mice inoculated with both CT26/HER2 tumor cells and CT26/p95HER2 cells. Interestingly, the same BVACs K1117 is the more proficient in activating the T cell immune response in both the un-typed and the HLA-A*0201, with a strong recruiting of HER2-specific IFN-g+ CD8+ T cells against ICD peptides and ECD peptides compared with the other two BAVC-HER2. Based on the previous results, the authors evaluated the synergistic effect between the HER2 K1117 antigen-based vaccine and immunotherapy against the checkpoint receptor PD-1. Interestingly, the combination showed the highest antitumor effect in BALB/c mice inoculated with CT26/HER2 cells.

The paper is based on a very hard, well conducted project design. The produced antigens are built following a smart backbone, and the idea of increasing the immunogenicity through the addiction of a partial ICD of HER2 seems to be correct. The final results concerning the combination of vaccine and anti-PD-1 are only preliminary but interesting. The statistics are correct, there are some typos and some images have to be improved. There are some points that have to be addressed:

Major issues

  • The author correctly stressed in the manuscripts the need of a non-oncogenic antigen as pivotal part of an inactive vaccine. They commented this main point in lines 364-372. The authors claim that the two, partial ICD-composed HER2 antigens lack the oncogenic ability because of the absence of the tyrosine kinase domain. This is theoretically correct, but they have no proof in principle. The authors could evaluate by using western blots the downstream pathway activation (for example AKT or ERK phosphorylation) in the THP-1 human monocytic cells used in the section 3.1, to confirm the inactivity of these antigens.
  • In the latter part, authors show results of combined K1117 HER2 antigen-based vaccine and anti-PD1 therapy, but they do not specify the moAb nor the posology of this drugs. This information should be provided.
  • In the context of immunogenicity and response to anti-PD1 therapy, Interferon gamma (IFN-γ) is recognized as an important biomarker to the response to immunotherapy in some tumor types (doi: 1177/1758834017749748) through the upregulation of the PD-L1 expression in tumor cells. Interestingly, the K1117 recruits IFN-g+ CD8+ T cells. Would it be possible to demonstrate a correlation between this T cell specific recruiting and the presence of PDL1 on tumor cells in the BALB/c mice inoculated with CT26/HER2 cells? IHC or other methods could be used.
  • The HER2 protein is a well-known oncogene in different disease, with a critical role in particular in about 15% of breast carcinomas (largely belonging to the HER2-enriched subtype at transcriptomic level). HER2 overexpression, mainly caused by ERBB2 gene amplification, leads to a very aggressive form of breast cancer, in which HER2 is target for different moAbs or TKIs. Several in vitro models for this BC subtype have been developed. Can the authors use or speculate on the use of a BC cell line with ERBB2 amplification for an inoculation in mice to evaluate the efficiency in the reduction of tumor growth?
  • Since the HER2 targeted agents are currently the only form of personalized therapy in HER2-enriched BC, do the authors can plan experiments of combination between the described BVACs and the anti-HER2 therapies?

Minor issues

  • Please add the ATCC certificate for CT26 and WEHI-164 (line 101)
  • Please add the reference sequence for HER2 in line 129
  • Please add the schematic project design for mouse analyses reported in figure 3, as for example figures 5 and 6.
  • Please add in figure 1B the ladder size
  • Please add in figure legend the explanation for the figure 1E
  • Please check and correct the panel letters for figure legend 2
  • Please correct the typo in line 330: INF- lacks of g

Reviewer 2 Report

The current study, although intriguing, needs a few issues to be addressed before it can be considered for publication. 

Line 46 - 49: 'Although B cells and monocytes ......... NKT cells and B cells/monocytes'. Please provide a references supporting this statement.

Line 204 - 205: 'mean fluorescence intensity ...... was higher....'. Current data does not show that. The data shows percent positive cells which is comparable between the three constructs. Please show MFI data to substantiate the statement. 

Please add details of the transduction protocol. How many viable cells are obtained? Also, are the flow cytometry plots gated on 'live' cells?

Fig 2A: Labels are too small to read.

Fig 2: Legend is inconsistent with figure labels, there is no mention of 'C' or 'D'. Please specify the correct figure labels in the legend.

Fig 2: For vaccination with BVAC cells, 2X106 cells were injected. How many of these were live cells? Was any dead cell exclusion done prior to injection? This is crucial as dead cells can act as potent immune activators and can skew the observed responses.

Fig 3: I have two issues with this figure: 1. Injection with the modified vaccines only delays the tumor growth, but does not totally abrogate it. 2. To really measure efficacy of the vaccines, I would like to see a survival curve analysis. Therefore, in light of the above two points, I would urge the authors to exercise caution in making sweeping conclusions about the efficacy of the vaccines.

In fact, my main concern with this manuscript is the lack of survival analysis. My strong recommendation would be to perform additional experiments of survival analysis as a measure of efficacy.

Reviewer 3 Report

In this study, Jeon et al. analyze the immunogenicity of differently truncated forms of HER2 and their capacity to induce anti-tumor responses. This is tested in vivo in mice and in vitro with human blood mononuclear cells. The results identify a truncated HER2 with therapeutic potential for anti-tumor vaccination.

They show that in mice, HER2 K1117 produces immunogenicity comparable and sometimes lower than that of HER2 constructs K684 and K965 (Fig 2). However, differently than mice, HER2 K1117 better primes human PBMC in vitro to respond to intracellular domain HER2 peptides, than other HER2 constructs (Fig 5). Interestingly, the increase in immunogenicity relates to CD4 T and CD8 T cells. Moreover, the anti-tumor inducing capacity of K1117 based on HER2-bearing tumors is higher than that of the other constructs (Fig 3).

A problem of using tumor-associated antigens such as HER2 to activate the immune system against tumors is the toxicity associated with a potential attack on normal tissues. This point is not addressed in the study by Jeon et al. Indeed, in the mouse model used in this study this fact is not revealed, because the differences between human HER2 and its mouse ortholog probably spare the mouse normal epithelial tissues from immune attack initiated against CT26/HER2 tumors.

Then, whereas K1117 induces a strong response against HER2-expressing tumor, it is unclear whether human normal tissues expressing HER2 will be spared from the attack. This could be a potential issue in human therapy, taking into account the results of Fig 3 and 4: priming with K1117 shows increased effectiveness than K965 and K684 against CT26/HER2 tumors (Fig 3) than against CT26/p95HER2 tumors. This was unexpected, on the basis of results of Fig 5, because K1117 primes human PBMC to respond better to ICD HER2 peptides than to ECD HER2. This suggests that the potential use of K1117 as immunogen in humans could present side effects derived from normal epithelial tissues being under immune attack.

A further point to consider, is the amount of HER2 (and p95HER2) antigen expressed in CT26 tumors used as targets in this study. It could well be that the amount of HER2 in CT26 is high enough to raise an optimal cytolytic T cell response and anti-HER2 antibodies, but natural tumors might have a minor expression of HER2. Ideally, the authors could have carried out an in vitro study of the T cell response against a natural human HER2-positive tumor, but this requires access to patient samples what is not always possible.

Other points to ponder,

  • Line 27. It is stated: “In addition, BVAC-K1117 showed enhanced antitumor effects against truncated p95HER2-expressing CT26 tumors to BVAC-K965 and BVAC-K684 by inducing T cell responses against intracellular domain (ICD) epitopes.” This is not totally correct. Fig 4 shows no differences in the T cell response between K1117  and K965.
  • Fig 4A. Which is the target cell type loaded with ICD 15-mer peptides? This information is not provided in the main text nor in the figure legend.
  • In Line 317, the sentence reads “We used HLA-A2-restricted ECD 9-mer peptides (P369 and P435) and ICD 9-mer peptides (P689 and P971) for CD8+ T cell restimulation”. But CD4 T (and not CD8 T) is depicted in the ordinate axis of Fig 5G. Please, correct the ordinate axis of Fig 5G.

Also, in Figure legend 5, line 333 which reads: “After 11 days, the CD8 and CD4 T cells isolated from cocultured PBMCs… “ ; has to be modified to: “After 11 days, the CD8 T cells isolated from cocultured PBMCs …

  • Similarly, in line 319, the sentence reads “…and pooled 15-mer ECD peptides or pooled 15-mer ICD peptides for CD4+ T cell restimulation.” But CD8 T (and not CD4 T) is depicted in the ordinate axis of Fig 5H. Please, correct the ordinate axis of Fig 5H.

Minor points

  • Line 27. It is stated: “In addition, BVAC-K1117 showed enhanced antitumor effects against truncated p95HER2-expressing CT26 tumors to BVAC-K965 and BVAC-K684 by inducing T cell responses against intracellular domain (ICD) epitopes.” I surmise that the authors wanted to say the following: “In addition, BVAC-K1117 showed enhanced antitumor effects against truncated p95HER2-expressing CT26 tumors compared to BVAC-K965 and BVAC-K684 by inducing T cell responses against intracellular domain (ICD) epitopes.”
  • Fig 1E is not described in the figure legend nor in main text
  • Fig 2C and 2D are referred to in the figure legend as 2A and 2B
  • Line 238. It is stated: “All three BVACs expressing HER2 antigen-induced potent target cell 238 killing activity to BVAC without HER2 expression and BVAC-K965 was the most potent.” I assume that the authors wanted to say the following: “All three BVACs expressing HER2 antigen-induced potent target cell killing activity in comparison to BVAC without HER2 expression and BVAC-K965 was the most potent.” Please, revise.
  • Line 250. It is stated: “One, 5, 7, 8, 11, and 13 weeks after immunization, antibodies in the mouse sera were titrated by flow cytometry.” But the figure 2B shows sampling at 1, 5, 7, 9, 11 and 13 weeks.
  • Line 277. It is stated: “(B) CT26/HER2 tumor-bearing mice were administered BVACs with engineered antigens 4 time every week …“ Looking at the scheme of Fig 3B, I understand that the authors administered BVACs 4 times in total, but not 4 times each week. Please, correct the sentence.
  • Line 278. It is stated: “…every week when the tumor size reached 80 mm3 (C).” There is no figure 3C. Please delete this “C”.
  • Line 329. It is stated: “After 11 days, the CD8 and CD4 T cells were sorted from cocultured PBMCs and restimulated with a total HER2 peptide pool, and IFN-g production was measured with an ELISPOT assay (B, C and D).” But the 11 days do not fit to the scheme shown in Fig 5A, where peptide re-stimulation was at D7. Also, the “gamma” in “IFN-gamma” has not been printed in the pdf.
  • Line 353. It is stated: “The dose per treatment of BVAC was 2x106 cells per mouse and anti-PD-L1 antibody was 300 g per mouse.” The authors probably wanted to say: “The dose per treatment of BVAC was 2x106 cells per mouse and anti-PD-L1 antibody was 300microg per mouse.”
  • Line 193. The title of the paragraph states: 1. Development of adenoviral vectors carrying engineered HER2 antigens. Although this sentence has a clear meaning, the adenoviral vectors do not “carry” antigens, therefore, I would recommend modifying it to: Development of adenoviral vectors coding for engineered HER2 antigens.